# A Review of Evidence-Based Recommendations for Pericoronitis Management and a Systematic Review of Antibiotic Prescribing for Pericoronitis among Dentists: Inappropriate Pericoronitis Treatment Is a Critical Factor of Antibiotic Overuse in Dentistry

**DOI:** 10.3390/ijerph18136796

**Published:** 2021-06-24

**Authors:** Jan Schmidt, Martina Kunderova, Nela Pilbauerova, Martin Kapitan

**Affiliations:** Department of Dentistry, Charles University, Faculty of Medicine in Hradec Kralove and University Hospital Hradec Kralove, 500 05 Hradec Kralove, Czech Republic; Jan.Schmidt@lfhk.cuni.cz (J.S.); KunderoM@lfhk.cuni.cz (M.K.); KapitanM@lfhk.cuni.cz (M.K.)

**Keywords:** pericoronitis, antibiotics, dentistry, antibiotic resistance

## Abstract

This work provides a narrative review covering evidence-based recommendations for pericoronitis management (Part A) and a systematic review of antibiotic prescribing for pericoronitis from January 2000 to May 2021 (Part B). Part A presents the most recent, clinically significant, and evidence-based guidance for pericoronitis diagnosis and proper treatment recommending the local therapy over antibiotic prescribing, which should be reserved for severe conditions. The systematic review includes publications analyzing sets of patients treated for pericoronitis and questionnaires that identified dentists’ therapeutic approaches to pericoronitis. Questionnaires among dentists revealed that almost 75% of them prescribed antibiotics for pericoronitis, and pericoronitis was among the top 4 in the frequency of antibiotic use within the surveyed diagnoses and situations. Studies involving patients showed that antibiotics were prescribed to more than half of the patients with pericoronitis, and pericoronitis was among the top 2 in the frequency of antibiotic use within the monitored diagnoses and situations. The most prescribed antibiotics for pericoronitis were amoxicillin and metronidazole. The systematic review results show abundant and unnecessary use of antibiotics for pericoronitis and are in strong contrast to evidence-based recommendations summarized in the narrative review. Adherence of dental professionals to the recommendations presented in this work can help rapidly reduce the duration of pericoronitis, prevent its complications, and reduce the use of antibiotics and thus reduce its impact on patients’ quality of life, healthcare costs, and antimicrobial resistance development.

## 1. Introduction

Pericoronitis is a term referring to inflammation of the soft tissues around the crown of an erupting tooth or a tooth with incomplete eruption [1]. Although pericoronitis is a bacterial infectious disease, the cause is not primarily determined by the transmission of the infectious agent but by local morphological conditions. After the tooth has partially erupted into the oral cavity, the integrity of the dental follicle is breached, and the space between the tooth and the follicle is colonized by oral microflora. This newly formed ‘pocket-like’ area is difficult to keep clean, and bacterial plaque and debris tend to accumulate underneath the soft tissue cap. Such a confined space is predisposed to the development of inflammatory complications.

The incidence of pericoronitis is 4.92%, and 95% of cases occur with the lower third molar [2,3]. Although incomplete eruption can occur with any tooth, lower third molars are affected most frequently due to their localization. The highest incidence is in vicenarians (20–29 years old), corresponding with the average age of third molar eruption [2,4]. On the other hand, children and adults over the age of 40 years are rarely affected [2]. Several studies have reported the distribution of pericoronitis between the sexes to be insignificant with a slight female predominance [5,6,7,8]. The diagnosis is based on clinical examination and differential diagnosis. Early detection is the key to effective therapy based on local treatment; the application of antibiotics is reserved for severe cases where the spread of infection or systemic response is involved [9,10].

Antibiotics are commonly prescribed medications. As there is always a risk of developing resistance, antibiotics should be indicated only when necessary and after careful consideration. In 2019, The Global Alliance for Infections in Surgery had issued a 10-point recommendation of principles for appropriate antibiotic therapy in surgery, which includes [11]:Enhancing infection prevention and control;Controlling source control;Prescribing antibiotics when they are truly needed;Prescribing appropriate antibiotics;Prescribing antibiotics with appropriate dosage;Reassessing treatment when culture results are available;Using the shortest duration of antibiotics based on evidence;Educating staff;Supporting surveillance of antimicrobial resistance and healthcare-associated infections and monitoring of antibiotic consumption;Supporting an interdisciplinary approach.

Dentistry is a field in which the use of antibiotics can be effectively reduced by proper prevention. Regular preventive check-ups are thus essential for identifying problems and their solutions before the development of inflammatory complications. If inflammatory complications have already occurred, early therapy at the source usually prevents the spread of infection. If an antibiotic prescription is necessary, it is always advisable to take a sample for cultivation (if possible). Until the results of cultivation are known, antibiotics are prescribed based on empiric experience and evidence-based recommendations. Antibiotics should be prescribed at the appropriate dose concerning the minimum inhibitory concentration and necessary length of time. To meet these requirements, healthcare professionals must be regularly educated in prescribing antibiotics and should be aware that any application of antibiotics contributes to the development of resistance to these drugs. As antibiotics are one of the most prescribed drugs, their consumption needs to be reported and statistically evaluated at the regional, state, and global levels.

Around 842 antibiotic prescriptions per 1000 persons were made in the United States in 2011 and 836 in 2016 [12,13]. A study by Demirjan et al. revealed that 30–50% of prescribed antibiotics are either not necessary or not optimally prescribed [14]. A study by Koyuncuoglu et al. showed that in dentistry it is even more, and 96.6% of antibiotics prescribed by dentists are for irrational or uncertain indications [15]. Of all antibiotic prescriptions in the United States, 10% are prescribed by dentists; in Canada, it is 11.3%, and in France, it is 8% [12,16,17]. In other specializations, the prescription of antibiotics is reduced year to year; however, in dentistry, it is on the rise [16]. Studies on this topic agree that the cause of antibiotic overprescribing is insufficient information and education of dentists as well as the fear of litigation and diagnostic uncertainty [10,12,16]. To effectively address this issue, it is necessary to provide directions to healthcare professionals on appropriate and evidence-based therapy for situations in which antibiotics are unnecessarily prescribed in dentistry.

In this work, we focused on the treatment of pericoronitis. The treatment of this high incidence diagnosis usually does not require the use of antibiotics; however, several studies have identified pericoronitis as one of the main reasons for prescribing antibiotics in dentistry [17,18,19]. Additionally, the authors also emphasized the urge for better training of dentists in prescribing antibiotics and the need for practical guidelines of appropriate therapy [10,17,19,20,21]. Thus, this work aims to:(a)Provide the most recent, clinically significant, and evidence-based recommendations for pericoronitis diagnosis and proper treatment (Part A);(b)Systematically review antibiotic prescribing for pericoronitis (Part B).

## 2. Part A: Pericoronitis Evidence-Based Therapy

Efficient treatment of any disease, including pericoronitis, must be based on its understanding. This chapter provides an overview of pericoronitis, presenting the most recent, clinically significant, and evidence-based recommendations for its diagnosis and proper treatment.

### 2.1. Methods

The PICO strategy: The problem involved was the treatment of pericoronitis; the intervention was a summarization of therapeutic recommendations based on evidence; for comparison, Part B identifying antibiotic prescribing as a main therapeutical approach to pericoronitis was used; the outcome was an evidence-based guideline for the treatment of pericoronitis recommending the local therapy over antibiotic prescribing, which should be reserved for severe conditions. Thus, this narrative review aims to provide comprehensive guidance on pericoronitis management with emphasis on evidence-based therapy.

This problem was addressed by summarizing the current evidence-based recommendations of authorities in the field, such as professional societies, universities, government health agencies, and scientific articles, to review this topic.

### 2.2. Classification

Pericoronitis occurs around an imperfectly erupted tooth. During the tooth eruption, a transient presence of the operculum (see Table 1) is a natural part of the process, and in most cases, it regresses spontaneously after the tooth has reached complete functional contact with the antagonist. However, if the tooth does not erupt correctly, the operculum may persist, and the soft tissues around the tooth may be recurrently inflamed. Based on this, pericoronitis can be classified as transient or non-transient. Transient pericoronitis occurs during the tooth eruption and may be considered a complication of the teething process. Non-transient pericoronitis occurs after the end of tooth eruption.

Additionally, pericoronitis can be classified as acute or chronic. Acute pericoronitis manifests as a sudden and severe expression of inflammation signs—heat, pain, redness, swelling, and loss of function. Chronic pericoronitis displays as mild and protracted, while the manifestation of inflammation signs may be present but subclinical [9]. Chronic pericoronitis is one of the predispositions for acute form, and its shift from chronic to acute is referred to as acute exacerbation of chronic pericoronitis.

Classification of Pericoronitis:

In relation to tooth eruption process:Transient—occurs during the tooth eruption;Non-transient—occurs after the tooth eruption is terminated.

In relation to the development:Acute—sudden onset, severe symptoms;Chronic—protracted, mild or no symptoms.

Sometimes, pericoronitis is also called dentitio difficilis. However, this term generally reflects any problem associated with complicated tooth eruption and should be considered hypernym of pericoronitis (see Table 1).

### 2.3. Etiopathogenesis

Even though pericoronitis is a bacterial infectious disease, its cause is not primarily determined by the transmission of the infectious agent but by local morphological conditions. Microorganisms involved are mostly obligatory and facultative anaerobes, such as *Actinomyces, Prevotella, Veillonella, Micromonas*, or *Propionibacterium* spp.; however, aerobic species, like *Streptococcus* or *Staphylococcus*, are usually present as well [22,23]. These bacteria are commonly found even in the healthy oral cavity [24]. Thus, the problem is not simply the presence of these bacteria but their accumulation, overgrowth, and poor hygiene management in the confined space between the soft tissue and the tooth (Figure 1). From this perspective, pericoronitis may be considered a plaque-induced complication of tooth eruption.

The rotation and position of the tooth, the morphology of the flap, and the shape or size of the pocket and its orifice play an important role in pericoronitis development. A thick lobe covering most of the crown of a tooth, forming a deep space under the lobe with a narrow orifice, is a typical morphological predisposition to pericoronitis. The lower third molar position is usually evaluated by Pell and Gregory classification (Figure 2) and Winters’s classification (Figure 3). The results of studies on the mandibular third molar position and the occurrence of pericoronitis vary [8,25,26,27,28]. However, a meta-analysis on this topic conducted in 2019 revealed that:

There is no significant difference in the chance of pericoronitis between class I and II of Pell and Gregory classification;Third molars classified in position A had a greater chance of pericoronitis when compared to those in position B of Pell and Gregory classification;The vertical position of the lower third molar (Winter’s classification) is more associated with the occurrence of pericoronitis when compared to the other positions, while the horizontal position decreases the occurrence of pericoronitis [29].

In addition to the conditions necessary for the pericoronitis development, i.e., imperfectly erupted tooth and bacterial accumulation, we also recognize factors that contribute to its frequency and severity. They can be classified as local and systemic. The most common local factors include soft tissue trauma, poor oral hygiene, or foreign body entrapment. The pericoronal flap can be traumatized during mastication when it is irritated by food pressure or by the direct bite of an antagonist tooth. Traumatization may also be caused by inappropriate oral hygiene. Insufficient oral hygiene leads to retention of plaque around the flap and contributes to its inflammation or bacterial accumulation under it. Food scraps are among the most common foreign bodies trapped between the soft tissues and the tooth. Their subsequent decomposition promotes bacterial growth and elicits an inflammatory tissue response.

Systemic factors that facilitate the development of pericoronitis or worsen its course are generally all diseases and conditions that impair the immune system and wound healing. These factors may be temporary or permanent. Examples of temporary factors are mental and physical stress or upper respiratory tract infections [5]. Kay et al. demonstrated that the seasonal fluctuation in respiratory infection outbreaks paralleled the rises in patient numbers with pericoronitis, and 33% of patients (*n* = 2311) suffering from pericoronitis admitted to a preceding upper respiratory infection [5]. Very similar conclusions were published by Bataineh et al., reporting on upper respiratory infection to be the most predisposing factor for pericoronitis (37.9% of cases, *n* = 2151) [7]. Correspondingly, a study by Meurman et al. described a significant increase in the incidence of respiratory tract infection during the two weeks before acute pericoronitis with the highest occurrence three days before pericoronitis [30]. Stress was also identified to play a conspicuous role in pericoronitis, preceding it in 17–66% [2,5,7]. The general stress-related changes in the immune system may contribute to the exacerbation of pericoronitis; however, there are no mechanistic studies focused on the stress role in pericoronitis development [31]. Additionally, an interesting relationship to menstruation was observed. From women affected with pericoronitis, 4–12% admitted concurrent menstruation [5,7]. Furthermore, in Kay’s study, over half of the women with pericoronitis (*n* = 1202) were within a few days of the anticipated menstrual discharge [5]. The authors hypothesized that emotional aspects might play the main role, as most of these women admitted emotional symptoms characteristic of premenstrual tension. The premenstrual exacerbation was also reported in other diseases, e.g., asthma, systemic lupus erythematosus, or multiple sclerosis, and might be explained by fluctuations of immune cell numbers and function during the menstrual cycle [32,33]. It is possible that both factors, emotional stress and immune modulation during the premenstrual phase, may play a role in the exacerbation of pericoronitis. Examples of permanent systemic factors can be diabetes mellitus or immunodeficiency disorders. If pericoronitis manifests as an accompaniment to another disease, it can be classified as sequela to a compromised immune system, ergo as an opportunistic-like infection. Additional findings indicated an increased prevalence of pericoronitis in smokers [34]. Some studies suggested competitive sports activities are a risk factor for pericoronitis as its incidence in sport professionals was reported to be between 5–39% [35,36,37]. The authors do not identify sport as the cause, but nutritional, hygienic, and behavioral habits of professional athletes.

Local and systemic factors can appear in parallel and potentiate each other. For instance, an ongoing upper respiratory tract infection is a stress factor for the patient’s psyche and immune system. During the illness, the frequency of food intake and the quality of oral hygiene may decrease, leading to plaque accumulation. All these factors increase the risk of pericoronitis development. Causes and risk factors are summarized in Table 2.

### 2.4. Clinical Manifestation and Diagnosis

Pericoronitis manifests with the expression of inflammation signs—pain, redness, swelling, heat, and loss of function. Pain begins locally and is limited to the soft tissues around the erupting tooth. Patients usually describe it as pulsating and eventually radiating to the surrounding tissues and/or distant areas (soft palate, mouth floor, retromandibular and submandibular space, throat, ear, or temporomandibular joint) [38]. It usually worsens over time and becomes more accentuated under pressure on the affected area. Pain may also disturb sleep, and its exacerbation during mastication may limit food intake. During the clinical examination, swollen, reddened soft tissue above and around a tooth is found (Figure 4).

The magnitude of the edema and pain may prevent the patient from reaching the resting position of the mandible and force him/her to keep it in a depressed position. Traumatization or even ulceration of the soft tissues as well as purulent exudation may be observed. Tissue and detritus decomposition may cause malodorous breath (halitosis), bad taste, or changes in taste perception [39]. Regional submandibular and neck lymphadenopathy is usually unilateral. Bilateral lymphadenopathy, pyrexia, palatoglossal arch asymmetry, facial asymmetry, malaise, difficulty swallowing (dysphagia), or restriction in mouth opening, which may be accompanied by pain (trismus) are warning marks indicating a more severe course that may include infection spread to the adjacent tissue spaces, i.e., the deep spaces of head and neck [38,40].

Chronic pericoronitis displays mild and protracted symptoms, or its manifestation is subclinical. Periodic attacks of acute pericoronitis may refer to chronic pericoronitis with recurrent exacerbations.

Pericoronitis is diagnosed clinically based on the presence of soft tissue inflammation associated with the partially erupted tooth. In differential diagnosis, the following disorders should be considered: dental caries, pulpitis, pulp gangrene, periapical abscess, food packing, gingivitis, mucosal disorders, periodontal abscess, alveolar osteitis (dry socket), peritonsillar abscess, pterygomandibular space abscess, temporomandibular disorders, and myofascial pain. The detection of any of these diseases does not exclude the presence of pericoronitis. Performing an X-ray examination is advisable, especially in severe, persistent, or recurrent cases and when resistant to therapy.

### 2.5. Complications

Pericoronitis complications resulting to emergencies should be treated in a hospital. However, every dentist should be able to recognize it to prevent any delays in providing proper treatment.

Like any source of bacterial infection, pericoronitis is associated with the production of pus. If not evacuated, it accumulates, and an abscess is formed in the pericoronal space. Further accumulation of pus leads to its propagation. Local tissue structures, such as ligaments and preformed anatomical spaces, facilitate the progression of the infection into the surrounding areas like the sublingual space, submandibular space, parapharyngeal space, pterygomandibular space, infratemporal space, submasseteric space, and buccal space [41]. Pus collection behind the tonsil leads to the formation of the peritonsillar abscess, also known as quinsy. Its symptoms include fever, lymphadenopathy, throat pain, dysphagia, dyspnea, change of voice, and asymmetry of the palatal arch due to the pus collection [42]. Treatment is performed via pus evacuation, antibiotics, sufficient fluids, and pain medication [43]. Involvement of submandibular, sublingual, and submental spaces may lead to a life-threatening condition called Ludwig’s angina. Symptoms of this phlegmonous infection include swelling around the mandible and upper neck, fever, lymphadenopathy, throat pain, dysphagia, dyspnea, elevation of the mouth floor, and tongue displacement [44]. It manifests with an acute onset and spreads rapidly with a risk of airway obturation. Thus, early diagnosis and treatment are essential. The treatment follows the same principles—pus evacuation, antibiotics, sufficient fluids, and pain medication [45]. Successful treatment of these complications includes resolving the primary source of infection.

### 2.6. Therapy

The primary cause of pericoronitis is a morphological predisposition to the accumulation of bacteria, leading to inflammation of the surrounding soft tissues. The first phase of treatment focuses on the elimination of bacterial overgrowth and pain management. After the acute phase, it is necessary to ensure that it does not recur. Therefore, the accumulation of microbes must be prevented.

#### 2.6.1. Infection Management

Most cases of pericoronitis are resolved with local intervention, including debridement and irrigation of stagnation areas [46]. Antibiotics are reserved only for severe cases and when systemic symptoms are present.
A.Local intervention [47]
Irrigation of pericoronal space with a sterile solution (aqua pro injectione, saline, antiseptics for mucosa, e.g., hydrogen peroxide or chlorhexidine).Mechanical removal of plaque and debris (debridement) from the pocket using periodontal instruments and swabs gently.Irrigation and debridement may be combined to achieve better results.Any collection of pus should be drained.Traumatic occlusion, if present, should be prevented by soft tissue or occlusal adjustment. Extraction of antagonist tooth may be considered.The patient should be instructed in oral hygiene involving gentle and careful mechanical cleaning of the affected area and mouth rinsing with antiseptics (e.g., 0.12–0.2% chlorhexidine two times daily for 1 min).

Surgical intervention during the acute phase remains a controversial issue. Protagonists argue that this approach leads to a quick resolve. Antagonists consider it an unnecessary risk of spreading the infection. No satisfactory agreement has yet been reached on this issue [48]. If surgery is necessary, for instance, to drain the abscess, cautery or laser were shown to be more beneficial over scalpel [49,50]. Ozone therapy may be an adjunct to local therapy, but there is no evidence of its effectiveness [9]. Photodynamic therapy appears to be a promising adjunctive antibacterial therapy and is discussed separately. Local caustic agents such as chromic acid, phenol liquefactum, trichloroacetic acid, or Howe’s ammoniacal solution were used to chemical cauterization of the pain nerve endings [41]. However, the use of these toxic chemicals in the oral cavity is no longer encouraged [9]. Application of local anesthesia during the local intervention is possible, but its effectiveness may be reduced by the acidic environment of infected tissues [51]. Topical analgesics may be an alternative providing short-term pain relief long enough to perform a local intervention.B.Antibiotics

Indication: Adjunct to local treatment in infection spread or systemic involvement [52,53].

Prescription: Principles of appropriate antibiotic prescribing based on guidance by the Faculty of General Dental Practice in the United Kingdom issued in 2020 are shown in Table 3 [53].

Antibiotics are essential for the treatment of pericoronitis when the spread of infection or systemic involvement is present. In these cases, it is a vital indication. However, as any use of antibiotics contributes to the development of antibiotic resistance and reduces their further effectiveness, adherence to the principles for appropriate antibiotic therapy is necessary. Therefore, a sample of pus should be taken for microbial culture prior to antibiotics application. If necessary, antibiotics that exhibit the best efficacy, i.e., metronidazole and amoxicillin, should be used by the end of the culture results, as they are generally the most effective antimicrobials against anaerobic organisms causing oral infections [53,54,55]. Two systematic reviews suggested that there is no evidence to recommend one antimicrobial over another to manage odontogenic infections [53,54,56]. Antibiotics should be prescribed at the appropriate dose with respect to the minimum inhibitory concentration and the necessary length of time. Any divergence from these rules should be made only for solid reasons.

In severe cases, the frequency of application and/or dose can be increased, or consideration should be given to using both amoxicillin and metronidazole in combination. For patients who are allergic to penicillin, erythromycin may be used instead [9]. While taking metronidazole, the patient should be advised to avoid alcohol. The anticoagulant effect of warfarin might be enhanced by metronidazole. Patients with significant trismus, the swollen floor of the mouth, or difficult breathing must be transferred to the hospital.C.Photodynamic therapy

Antimicrobial photodynamic therapy (aPDT) is a cytotoxic non-invasive treatment option with a low tendency to induce drug resistance [57]. Briefly, this method includes an application of a photosensitizing agent in the target tissue and its activation by laser light of a specific wavelength in the presence of oxygen [58,59]. Upon irradiation, the photosensitizer molecules undergo excitation transferring energy to the oxygen molecule that consequently forms oxygen free radicals [60,61]. These free radicals are highly cytotoxic and help to eliminate bacteria [61,62]. This method is also used in dentistry, including the therapy of pericoronitis [59].

A clinical study by Corrêa et al. showed that aPDT combined with local mechanical intervention exhibited a statistically significant reduction in pathogens in periodontal pockets compared to sole local mechanical intervention [63]. Elsadek et al. compared sole debridement to debridement combined with adjunctive aPDT in pericoronitis. The combination significantly lowered TNF-α concentration in gingival fluid collected from around inflamed pericoronal flap and significantly reduced periodontal pathogens in plaque from the pericoronal pocket [61]. However, there was no significant effect on the pain scale. A study by Eroglu et al. demonstrated that aPDT adjunctive to amoxicillin prescription significantly lowered the presence of inflammatory cells in inflamed pericoronal tissues [64]. The authors concluded that a combination of antibiotics and aPDT showed superior histological and clinical outcomes than antibiotics alone.

Although the current number of studies focused on aPDT and pericoronitis is insufficient to make firm conclusions, this method appears to be a promising adjunctive antibacterial therapy for pericoronitis [64].D.Follow-up to check the effectiveness of treatment

#### 2.6.2. Pain Management

Pain, a symptom of inflammation, is the most common reason leading a patient suffering from pericoronitis to oral healthcare providers. It significantly reduces the quality of life and limits the patient in his/her daily routine, social life, eating a regular diet, chewing food, and talking [65]. Thus, pain relief should be an integral part of pericoronitis treatment. The analgesics of choice should be nonsteroidal anti-inflammatory drugs (NSAIDs) [66]. Whether by administering local anesthesia or topical anesthesia, pain management is also an essential part of local treatment as it increases patient compliance during the procedure.A.Oral analgesics

Indication: Pain that reduces the quality of life and limits the patient in the daily routine. Individually variable based on the subjective perception of pain.

Prescription: Principles of appropriate analgesic prescribing based on Drug Prescribing For Dentistry Dental Clinical Guidance issued by the National Dental Advisory Committee in partnership with National Health Service Education for Scotland issued in 2016 are summarized in Table 4 [67].B.Topical analgesics

Indication: Pain management during the local intervention. Due to its short duration and high concentration, topical analgesics should not be used for continuous pain relief. The indication before a meal is controversial as a potent analgesic effect can lead to unintentional self-inflicted damage.

Prescription: Assessment of the current literature did not provide any comprehensive guideline for the application of topical analgesics in the oral cavity. The overview based on the literature review is shown in Table 5.

The smallest possible doses of topical analgesics should be administered to prevent intoxication, and the application should be targeted only to the affected tissue. The area of application should be dry to facilitate absorption, and excess analgesics should be removed. Due to spray scattering, spray analgesics should first be absorbed into a swab which is then applied to the affected tissue to minimize the dose.

#### 2.6.3. Prevention

Prevention of disease recurrence is one of the critical factors of effective treatment and antibiotic use reduction. The cause of pericoronitis is the accumulation of microbes due to local morphological conditions. Therefore, the only successful prevention of pericoronitis is the prevention of bacterial stagnation. Although the desired outcome of treatment is clear, the way to achieve it may be a therapeutic dilemma.A.Tooth extraction

The main query usually addresses the need for extraction. Guidelines and recommendations can contribute to finding the answer; however, given the complexity of the problem and the need for an individual approach to each case, the final decision will always require the insight of a dental specialist. Regarding the extraction of lower third molar due to pericoronitis, guidelines for clinical practice in the NHS issued by NICE provide these recommendations: 


*Specific attention is drawn to plaque formation and pericoronitis. Plaque formation is a risk factor but is not in itself an indication for surgery. The degree to which the severity or recurrence rate of pericoronitis should influence the decision for surgical removal of a third molar remains unclear. The evidence suggests that a first episode of pericoronitis, unless particularly severe, should not be considered an indication for surgery. Second or subsequent episodes should be considered the appropriate indication for surgery.*


The decision should also reflect whether further tooth eruption can reverse the current adverse morphological conditions and whether there is a chance to achieve a functional tooth position. Additional factors, such as autotransplantation, orthodontic treatment, the proximity of the mandibular canal, and disorder and medical history of the patient, should also be evaluated. If the decision is made to extract the tooth as a definitive solution to pericoronitis, it should not be unnecessarily postponed. Teeth with incomplete development indicated for removal should be extracted without undue delay to minimize invasiveness as well as bone loss and eventual complications [73].B.Pericoronal tissue surgery

An alternative to tooth extraction is pericoronal tissue surgery. This includes removing soft tissue covering the tooth, i.e., operculectomy, and eventually gingivoplasty around the tooth to eliminate deep pockets [74]. This can be achieved by a conventional procedure using a scalpel; however, more advanced methods using a diode laser or cautery have been shown to provide relevant benefits. Laser or cautery gingivectomy are safe procedures performed to remove excess soft tissue and expose the crown of the partially erupted teeth, allowing maintenance of an improved level of hygiene [75]. Laser and cautery use in pericoronal tissue management is associated with minor bleeding, suturing, postoperative pain, and complications than scalpel [49,74,76]. The result of the treatment should be the elimination of all abundant tissues that contribute to the retention of bacteria and do not allow its removal during standard dental hygiene. An examination should follow treatment to assess the outcome. If the desired result is not achieved, further soft tissue surgery should be considered. Compared to extraction, pericoronal tissue surgery is followed by less pain and complications [77]. In some cases, further adjunct use of orthodontics may help to achieve proper tooth position and alleviate the problem [75].C.Oral hygiene

Careful oral hygiene is an essential part of any plaque-related disease prevention, including pericoronitis, and all patients should be instructed so [78].

### 2.7. Discussion

Pericoronitis is a common disease with an incidence of around 5%. It occurs mainly in people between 20–29 years, severely affecting their daily routine, social life, eating a regular diet, chewing food, and talking. Its cure is quick, easy, cheap, and with no need for systemic antibiotic application if detected early and appropriately treated. However, the reality is different, and the systematic review (Part B) showed that a large proportion of dentists routinely prescribe antibiotics for pericoronitis. As professionals in the field, we deal with the consequences of neglected or inadequately treated pericoronitis on a daily basis. We consider the following to be the main reasons preventing an effective and appropriate approach to the treatment of pericoronitis:The diagnosis of pericoronitis is late due to failure to seek a medical examination or poor diagnosis. This contributes to the development of infection spread and systemic symptoms.Improper treatment of pericoronitis can worsen the patient’s condition due to the treatment itself and delaying the proper care.Ignorance of principles for appropriate antibiotic therapy.

Timely and adequate treatment is essential in pericoronitis management. Early diagnosis and proper therapy save the patient from pain and complications, reduce the patient’s social and labor indisposition, eliminate or lessen the need for antibiotic prescribing, thus decreasing the use of antibiotics and the risk of resistance development. These aspects, accentuated by the frequency of pericoronitis, are solid arguments for raising awareness of this disease, especially its efficient treatment.

Therefore, we decided to summarize the current evidence-based recommendations of authorities in the field, such as professional societies, universities, government health agencies, and scientific articles, to comprehensively review this topic. Emphasis was placed on clinical importance with a broad overview of current trends in pericoronitis surgical and pharmacological management, including local intervention, antibiotics prescription, pain management, and prevention. Adherence to these rules can help rapidly reduce the duration of the disease, prevent its complications, minimize the use of antibiotics, and thus reduce its impact on patients’ quality of life, healthcare costs, and antimicrobial resistance development.

### 2.8. Conclusions

Pericoronitis is a common complication of tooth eruption, which reduces patients’ quality of life and, if neglected, can lead to a life-threatening condition. Its early detection and treatment lead to faster recovery, prevention of complications, decreased antibiotic use, and savings in healthcare costs. This chapter summarizes current knowledge of pericoronitis classification, etiopathogenesis, symptoms, diagnosis, treatment, and prevention, emphasizing clinical significance.

The second aim of this work is to provide a systematic review of antibiotic prescribing for pericoronitis among dentists. Its results are in striking contrast to Part A of this work and accentuate the need for better adherence of dentists to pericoronitis evidence-based therapy.

## 3. Part B: Systematic Review of Antibiotic Prescribing for Pericoronitis

### 3.1. Methods

For the question formulation, the PICO strategy (problem; intervention; comparison; outcome) was used. The problem involved was the treatment of pericoronitis; the intervention was the antibiotic prescribing for pericoronitis treatment; treatment of pericoronitis not involving antibiotic prescribing was used as a comparison; the frequency of antibiotic prescribing in the overall treatment of pericoronitis was the outcome of interest. Thus, systematic review aims to clarify the frequency of antibiotic prescribing for pericoronitis.

This problem can be addressed by analyzing sets of patients treated for pericoronitis or analyzing questionnaires that identify dentists’ therapeutic approaches to pericoronitis. Both of these options were included in this systematic review and analyzed separately.

#### 3.1.1. Eligibility

Original articles and scientific research reports that reported on the use of antibiotics for pericoronitis treatment were included in this review. Studies and scientific research reports that did not distinguish between antibiotic treatment and non-antibiotic treatment of pericoronitis, studies with less than 15 participants, in vitro studies, review articles, conference summaries, letters to the editor, and case reports were excluded.

#### 3.1.2. Search Strategy

The following databases were searched: Web of Science and Medline without language restriction and with time limitation from January 2000 to May 2021. Relevant records were identified using the following search terms. The terms used in the search were the following keywords, according to the MeSH (Medical Subject Heading): Pericoronitis AND Antibiotics. After duplicate removal, the titles and abstracts of the articles found were read independently by three of the authors (J.S., N.P., M.K. (Martin Kapitan)). The studies potentially meeting this review’s inclusion criteria were identified and then independently assessed. Discrepancies were resolved by consensus.

#### 3.1.3. Data Extraction

Data were extracted from the publications by three authors (J.S., N.P., M.K. (Martin Kapitan)) independently. Disagreements were resolved by consensus. The main interest was the treatment of pericoronitis with and without antibiotics. Additional information, i.e., author, yea and country of study, type of antibiotics used, and frequency of antibiotic use within the surveyed diagnoses and situations, were also collected.

### 3.2. Results

#### 3.2.1. Study Selection

A total of 65 potentially relevant records were identified and further processed (Figure 5). Additional 5 records were identified through other sources. After that, duplicate removal was performed, and 56 records were further examined based on the title and abstract. Then, 19 records were removed as they did not cover the eligibility criteria included in Appendix A. A total of 37 articles were identified to be full-text read. Thereafter, 26 articles were excluded due to the reasons described in Appendix A. Finally, a total of 11 studies were included in the present review.

#### 3.2.2. Study Characteristics

All studies included in the systematic review evaluated therapeutic approaches to pericoronitis based on the use of antibiotics. Out of these studies, 6 were questionnaires that identified dentists’ therapeutic approaches to pericoronitis, and 5 were studies with sets of patients treated for pericoronitis. These two categories were evaluated separately.

#### 3.2.3. Questionnaires among Dentists

Six studies based on self-administered questionnaires among dentists identifying their therapeutic approaches to dental-related diagnoses and situations, including pericoronitis, were evaluated, and the summarization of their approaches to pericoronitis is presented in Table 6 [10,17,20,21,79,80]. The position of pericoronitis in the frequency of antibiotic used for its treatment within the surveyed diagnoses and situations is presented in Table 7. The frequency of antibiotic types used in the studies is in Table 8. The evaluation of antibiotic prescribing and evaluation of the need for further education according to the studies’ authors is presented in Table 9.

#### 3.2.4. Studies Involving Patients Treated for Pericoronitis

Five studies reporting on the use of antibiotics in the treatment of patients with pericoronitis were evaluated, and their summarization is presented in Table 10 [18,19,81,82,83]. The position of pericoronitis in the frequency of antibiotic use for its treatment within the surveyed diagnoses and situations is presented in Table 11. The frequency of antibiotic types used in the studies is in Table 12. The evaluation of antibiotic prescribing and evaluation of the need for further education according to the authors of the studies is presented in Table 13.

### 3.3. Discussion

Antibacterial therapy is one of the main achievements of 20th century medicine, which influenced the development of human society. As antibiotic resistance rises, it becomes one of the main issues of contemporary medicine and has also been identified as one of the major global challenges for the 21st century [84,85]. Indeed, less than a hundred years after its discovery, the effectiveness of antibiotics is declining, and the development of new molecules struggles. The growth of resistant strains increases morbidity, mortality, and healthcare costs and is a threat to public health. The development of new drugs may counter these adverse clinical and economic outcomes; however, this solution is highly unpredictable and risky to rely on. Thus, any healthcare professional should also contribute to the fight against antibiotic resistance by following the principles of appropriate antibiotic therapy.

Dentistry contributes significantly to the overall use of antibiotics, but only a minority of dentists prescribe antibiotics appropriately [15,86]. In contrast to most other medical fields, it is dentistry where antibiotic prescription does not decrease but increases [16]. Studies on this topic emphasize the need for the education of dentists in appropriate antibiotic therapy; however, they usually do not provide any comprehensive recommendations and practical directions.

This systematic analysis of antibiotic prescribing for pericoronitis revealed that in all reviewed studies, antibiotics were the leading therapeutic choice for pericoronitis treatment, except the studies by Combes and Mahmoody. Combes et al. evaluated dental care provision to UK military personnel, and less use of antibiotics could be due to better adherence to appropriate prescribing among dentists serving in the military compared to their civilian counterparts. For the study of Mahmoody, the reasons for lower antibiotic prescribing for pericoronitis in Germany were not identified. In studies evaluating multiple diagnoses and not only pericoronitis, it was also interesting to observe a comparison of antibiotic prescribing between diagnoses. For instance, in the study by Combes, antibiotics prescribed for pericoronitis in the UK and Afghanistan accounted for 54.3% and 64.9% of all prescribed antibiotics, respectively. In the study by Cope, it was 20.6%. However, these values are influenced by the absolute frequency of patients with a specific diagnosis. Therefore, it is more appropriate to compare the frequency of antibiotic prescriptions in treating specific diagnoses. In all questionnaires, which also included dental diagnoses and conditions other than pericoronitis, pericoronitis was among the top 4 in the frequency of antibiotic use. In studies involving patients, it was even higher as pericoronitis was first or second. This incoherency may be due to different diagnoses and conditions involved in the questionaries and in the studies with patients. In the questionaries, specific conditions requiring antibiotic therapy were included (e.g., prophylaxis before surgery). On the other hand, the studies including patients evaluated mainly the clinical diagnoses for which the patients were treated. Therefore, studies involving patients better reflect the real situation when comparing the position of pericoronitis in the frequency of antibiotic use. These results show that pericoronitis contributes notably to the consumption of antibiotics in dental medicine. The most commonly prescribed antibiotics for pericoronitis were amoxicillin and metronidazole, which is consistent with the appropriate choice of antibiotics for this diagnosis [53].

The general evaluation of the study results by their authors was very critical of the therapeutic approaches of dentists, especially when it came to prescribing antibiotics. In reality, antibiotics were often prescribed arbitrarily and unnecessarily. For illustration, in the study by Palmer, almost half of the dental practitioners surveyed used antibiotics due to uncertainty about the diagnosis (47.3%) or when under the pressure of time (30%). The situation where treatment had to be delayed accounted for 72.5% of prescribing. In the study by Salako, the respondents considered the following reasons to be justified in prescribing antibiotics: postponement of specific treatment (42.3%), diagnosis not certain (20.2%), patient’s social background (14.3%), convenience (7.7%), patient’s expectations for a prescription (4.2%). On the other hand, dentists were aware of the development of bacterial resistance resulting from the use of antibiotics and considered education in this issue to be important. For example, in the study by Baudet, 91% of respondents replied that antibiotic resistance is of concern, and 47.7% felt inadequately informed and trained regarding antibiotic use. A total of 93% of them wished to receive updates of guidelines in the practical form.

Most authors indicated the need for further education of dentists in appropriate antibiotic therapy. The authors also concluded that most dental practitioners prescribe antibiotics for pericoronitis, although it can be effectively treated by local measures [10,17,19,21]. To reverse it, the authors emphasize the need for an evidence-based standard of care [10,17,19,20,21].

### 3.4. Conclusions

This systematic review of antibiotic prescribing for pericoronitis shows an abundant and unnecessary use of antibiotics that is in striking contrast to the evidence-based recommendations described in Part A of this work. Questionnaires among dentists revealed that almost 75% of them prescribed antibiotics for pericoronitis, and pericoronitis was among the top 4 in the frequency of antibiotic use within the surveyed diagnoses and situations. Studies involving patients showed that antibiotics were prescribed to more than half of the patients with pericoronitis, and it was one of the top 2 diagnoses and situations in the frequency of antibiotic use. The most prescribed antibiotics for pericoronitis were amoxicillin and metronidazole. The use of antibiotics should be reserved for severe cases of pericoronitis where the spread of infection or systemic response is involved. As early diagnosis and proper treatment of pericoronitis do not require a prescription of antibiotics, the abovementioned findings show the overuse of antibiotics and demonstrate the need for further education of dentists in pericoronitis therapy as well as in principles of appropriate antibiotic therapy described in Part A.

## Figures and Tables

**Figure 1 ijerph-18-06796-f001:**
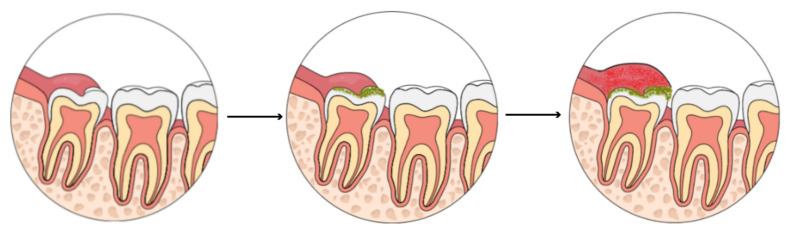
Pericoronitis development—illustrative scheme. Images demonstrate a partially erupted third molar covered by soft tissue. Plaque and detritus (green) stagnation lead to soft tissue inflammation.

**Figure 2 ijerph-18-06796-f002:**
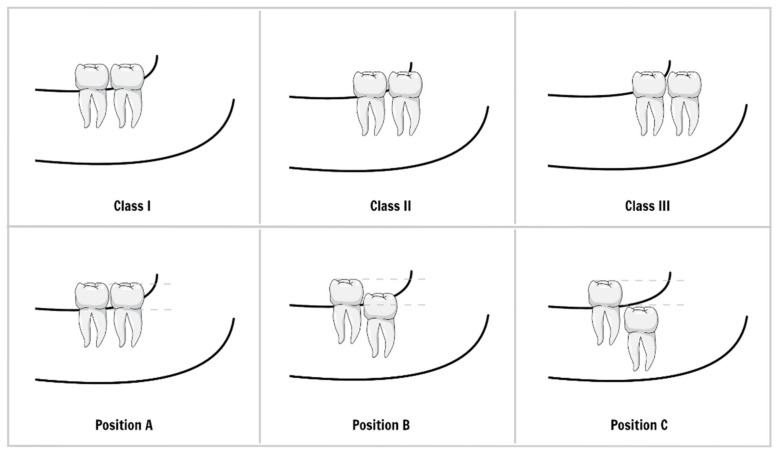
Pell and Gregory classification. The classes are based on the relationship between the lower wisdom tooth (third molar) and the mandible ramus. The positions are based on the vertical relationship between the second and third molars.

**Figure 3 ijerph-18-06796-f003:**
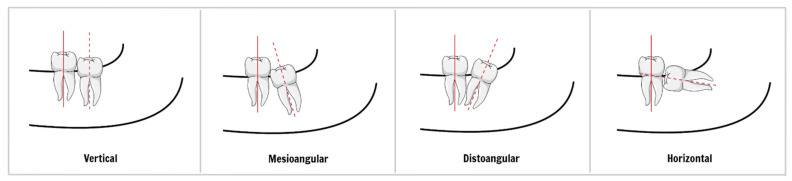
Winters’s classification. The classification is based on the inclination of the impacted wisdom tooth (third molar) to the long axis of the second molar.

**Figure 4 ijerph-18-06796-f004:**
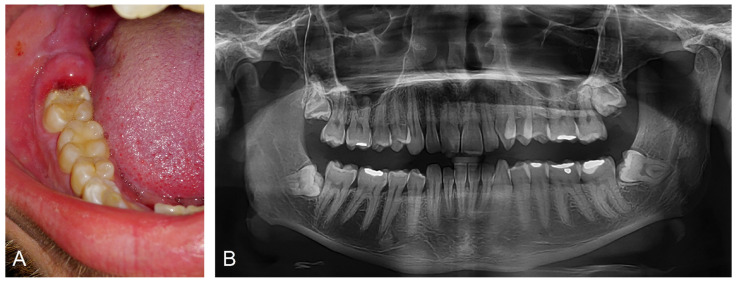
Clinical manifestation of pericoronitis and panoramic radiograph. Image (**A**) demonstrates inflamed soft tissues covering incompletely erupted right lower third molar. Image (**B**) displays a preoperative panoramic radiograph of the same patient demonstrating incompletely erupted third molars.

**Figure 5 ijerph-18-06796-f005:**
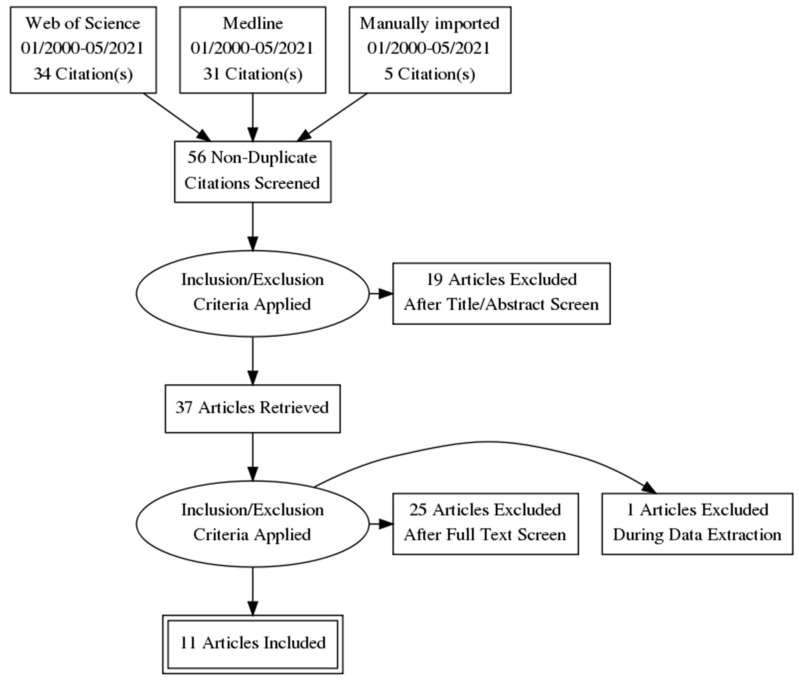
Flow diagram. A total of 65 potentially relevant records were identified searching Web of Science and Medline and further processed. Additional 5 records were identified through other sources. After duplicate removal, 56 records were further examined based on title and abstract. Then, 19 records were removed as they did not cover the eligibility criteria (described in Appendix A). A total of 37 articles were identified to be full-text read. Thereafter, 26 articles were excluded due to the reasons described in Appendix A. Finally, a total of 11 studies were included in the present review.

**Table 1 ijerph-18-06796-t001:** Terminology. Table 1 explains clinical terms associated with inflammation of soft tissues around erupting or imperfectly erupted teeth, etymology, and semantic relations.

**Term**	**Meaning**	Etymology	Semantic Relation
Operculitis	inflammation of operculum; operculum is a clinical term for the soft tissue covering a partially erupted tooth	Latin verb *operire* ‘to cover’	Hyponym of pericoronitis
Pericoronitis	inflammation of the tissues around the tooth crown	Greek prefix *peri-* ‘around’ Latin noun *corona* ‘crown’ Greek suffix *-itis* ‘inflammation of a tissue’	Hypernym of operculitis Hyponym of dentitio difficilis
Dentitio difficilis	difficult teething	Latin verb *dentitio* ‘teethe’ Latin adjective *difficilis* ‘difficult’	Hypernym of pericoronitis

**Table 2 ijerph-18-06796-t002:** Causes and risk factors for pericoronitis.

Causes	Risk Factors	
	Local	Systemic
	Pericoronitis in anamnesis	Upper respiratory tract infection
Imperfectly erupted tooth	Poor oral hygiene and plaque retention	Mental or physical stress
Bacterial accumulation	Traumatization of pericoronal soft tissues	Diseases impairing the immune system or wound healing (diabetes mellitus)
	Debris entrapment	Premenstrual phase
		Smoking

**Table 3 ijerph-18-06796-t003:** Antibiotic prescription for pericoronitis. Symbols (*) indicate further notes. Notes are provided within the table.

**Metronidazole**			
		**Adults**	**Children** (over 10 years)
Orally		400 mg *	200–250 mg *
Intravenously		500 mg **	7.5 mg/kg ***

Notes:	*	three times daily for up to five days	
	**	every 8 h given over 20 min	
	***	every 8 h (max. 500 mg per dose)	
**Amoxicillin**			
		**Adults**	**Children** (over 12 years)
Orally		500 mg *	500 mg ***
Intravenously		500 mg **	–

Notes:	*	every 8 h for up to five days; 1 g every 8 h in severe infection	
	**	every 8 h; 1 g every 6 h in severe infection	
	***	every 8 h; 1 g every 8 h in severe infection	

**Table 4 ijerph-18-06796-t004:** Oral analgesic prescription for dentistry.

**Ibuprofen**		
**Adults**	**Children**	
400 mg	6–11 months	50 mg
	1–3 years	100 mg
	4–6 years	150 mg
	7–9 years	200 mg
	10–11 years	300 mg
	12–17 years	300–400 mg

Notes:	The doses can be used four times a day for up to five days In adults, the dose can be increased to a maximum of 2.4 g daily	
	Administration preferably after food	
**Aspirin**		
**Adults**	**Children**	
600 mg	<16 years	– *
	>16 years	as for adults

Notes:	The doses can be used four times a day for up to five days Blood thinner Aspirin should not be prescribed after or before surgery	
	Administration preferably after food * not recommended for children due to Reye’s syndrome	
**Diclofenac**		
**Adults**	**Children**	
50 mg	– *	

Notes:	The doses can be used three times a day for up to five days The maximal daily dose is 150 mg * not recommended for dental use in children	

**Table 5 ijerph-18-06796-t005:** List of topical analgesics, their availability, onset time, and duration.

Topical Analgesics		Availability	Concentration	Onset Time (min)	Duration (min)
**Benzocaine** * [68,69]		gel, spray, ointment, solution	1–20%	0.5	5–15
**Tetracaine Hydrochloride** ** [69]		spray, ointment, solution	0.2–2.0%	2	20–60
**Lidocaine** [69,70]		gel, spray, ointment, solution	2–5%	1–2	15
**Cetacaine** [69]		solution	14% benzocaine	0.5	30–60
			2% butamben		
			2% tetracaine-		
			hydrochloric acid		
**EMLA** *** [69,70,71]		cream	1:1 mixture of	2	10
			2.5% prilocaine and 2.5% lidocaine		
**Oraqix** [69,72]		gel	2.5% lidocaine and 2.5% prilocaine	0.5	20
Notes:	*	risks: cross allergies to PABA and ester-type anesthetics; methemoglobinemia			
	**	quickly absorbed into the mucosa, dose limitation is 20 mg per session in healthy adults			
	***	eutectic mixture of local anesthetics			

**Table 6 ijerph-18-06796-t006:** Questionnaires among dentists regarding the antibiotic prescribing for pericoronitis.

Author	Year	Country	Question Design	Number of Respondents (*n*)	Outcome		(*n*)/(On) in %
					Specification	Number (On)	
Baudet	2020	France	Situation (pericoronitis) in which antibiotics were reported to be prescribed to a healthy patient.	408	out of (*n*)	239	58.6
Wehr	2019	Texas, USA	An emergency treatment preferred for acute pericoronitis involved antibiotics.	72	out of (*n*)	41	56.9
Ramadan	2019	Sudan	Pericoronitis is an indication for antibiotic prescribing.	100	yes	77	77.0
Vessal	2011	Iran	Dental practitioners that would prescribe antibiotics for pericoronitis.	219	ouf of (*n*)	147	67.1
Salako	2004	Kuwait	Should antibiotics be prescribed for pericoronitis?	168	yes	122	72.6
Palmer	2000	UK	Dental practitioners prescribing antibiotics for pericoronitis.	929	out of (*n*)	780	84.0
**Total**				**1896**		**1406**	**74.2**

**Table 7 ijerph-18-06796-t007:** The position of pericoronitis in the frequency of antibiotic use for its treatment within the surveyed diagnoses and situations.

Author	Country	The Position of Pericoronitis in the Frequency of Antibiotic Use for Its Treatment within the Surveyed Diagnoses and Situations (*n*)	(*n*)
Baudet	France	3rd	5
Wehr	Texas, USA	not specified	not specified
Ramadan	Sudan	4nd	9
Vessal	Iran	3rd	10
Salako	Kuwait	3rd	8
Palmer	UK	2nd	15

**Table 8 ijerph-18-06796-t008:** Frequency of prescribed antibiotics.

Author	Country	Frequency of Prescribed Antibiotics	%
Baudet **	France	amoxicillin spiramycin + metronidazole combination amoxicillin-clavulanic acid	65.8 11.6 10.3
Wehr	Texas, USA	not specified	
Ramadan **	Sudan	metronidazole amoxicillin amoxicillin-clavulanic acid	35.0 31.4 17.4
Vessal	Iran	not specified	
Salako *	Kuwait	amoxicillin metronidazole Penicillin	68.7 13.0 10.4
Palmer *	UK	metronidazole amoxicillin penicillin	67 30 10
Note:	*	Data for pericoronitis-related prescription	
	**	Data for all dental-related prescription, including pericoronitis	

**Table 9 ijerph-18-06796-t009:** Authors’ general evaluation of treatments reported by respondents of their studies and authors’ opinion on the need for further education in appropriate therapy.

Author	Authors’ General Evaluation of Treatments Reported by Respondents	Need for Further Education
Baudet	This nationwide study… shows the same trend as in other countries in terms of high prevalence of misuse and overuse of antibiotics.	Yes
Wehr	Skewed reasoning for treating pericoronitis.	Yes
Ramadan	Shortfalls in the knowledge of the participants regarding clinical indications and choice of antibiotic.	Yes
Vessal	Unfortunately, more than 60% of our dental practitioners surveyed would prescribe antibiotics routinely for periodontal abscess and pericoronitis. Most of those surveyed used antibiotics routinely for conditions where local treatment would be sufficient. Our findings indicate that the scientific basis for prescribing antimicrobial agents was neglected by the majority of the respondents.	Yes
Salako	The results of this study have demonstrated the lack of consistency in the rationale use of antibiotics.	Yes
Palmer	This survey supports the conclusion that there is overprescribing of antibiotics.	Yes

**Table 10 ijerph-18-06796-t010:** Antibiotic prescription for patients with pericoronitis.

Author	Year	Study Type	Country	Number of Patients Treated for Pericoronitis (*n*)	Out of (*n*), Antibiotics Prescribed (An)	(An)/(*n*) in %
Combes *	2019	prospective	UK	69	26	37.7
			Afghanistan	478	183	38.3
Bjelovucic	2019	retrospective	Croatia	406	261	64.3
Mahmoodi	2015	retrospective	Germany	119	44	37.0
Cope	2016	cross-sectional	UK	72	67	93.1
Tulip	2008	retrospective	UK	46	39	84.8
**Total**				**1190**	**620**	**52.1**

Note:	*	Dental care provision to UK military personnel serving in Afghanistan and at UK military home bases				

**Table 11 ijerph-18-06796-t011:** The position of pericoronitis in the frequency of antibiotic use for its treatment within the surveyed diagnoses and situations.

Author	Country	The Position of Pericoronitis in the Frequency of Antibiotic Use for Its Treatment within the Surveyed Diagnoses and Situations (*n*)	(*n*)
Combes	UK	1st	≥8 *
	Afghanistan	1st	≥8 *
Bjelovucic	Croatia	2nd	10
Tulip	UK	2nd	14
Cope	UK	1st	9
Mahmoodi	Germany	2nd	5

Note:	*	Total number of all diagnoses not clearly specified	

**Table 12 ijerph-18-06796-t012:** Frequency of prescribed antibiotics.

Author	Country	Frequency of Prescribed Antibiotics	%
Combes	UK	not specified	
	Afghanistan		
Bjelovucic **	Croatia	penicillin + clavulanic acid	70.5
		clindamycin	13.0
		metronidazole + penicillin	7.2
Tulip **	UK	amoxicillin	45.6
		metronidazole	32.3
Cope	UK	not specified	
Mahmoodi *	Germany	amoxicillin	21.8
		amoxicillin + clavulanic acid	10.9
		clindamycin	3.4
Notes:	*	Data for pericoronitis-related prescription	
	**	Data for all dental-related prescription	

**Table 13 ijerph-18-06796-t013:** Authors’ general evaluation of treatments reported in their studies and authors’ opinion on the need for further education in appropriate therapy.

Author	Authors’ General Evaluation of Reported Treatments		Need for Further Education
Combes	It could be argued that treatment of UK military personnel is predominantly more operative than their civilian counterparts… with reduced reliance on antibiotic therapy for the management of pericoronitis.		not stated
Bjelovucic	Antibiotics were occasionally prescribed without dental treatment, namely in pericoronitis (46.3%). Multiple possible issues in the prescription of antibiotics were observed, ranging from administration for inappropriate indications to noncritical and excessive prescription.		yes
Mahmoodi	Compared to the literature, surgical or dental interventions were more often performed with a more restrictive use of antibiotics.		not stated
Cope	The current study demonstrated high levels of guideline-incongruent antibiotic prescribing by general dentist practitioners. Cases of pericoronitis, apical abscesses and acute periodontal conditions account for over 70% of all antibiotics prescribed (20.6% for pericoronitis).		yes
Tulip	The study has highlighted that many GDPs are not familiar with current clinical and best practice guidelines on patient examination, management with respect to the correct prescribing of antibiotics for dental infections.		not stated *
Note:	*	The authors stated that the reasons why dentists failed to provide definitive treatment and the high number of prescriptions for antibiotics require further research.	

## Data Availability

The data presented in this study are openly available in Zenodo at https://zenodo.org/record/5014844#.YNRLeugza70 (accessed on 22 June 2021).

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
