# Peer review of "A Review of Evidence-Based Recommendations for Pericoronitis Management and a Systematic Review of Antibiotic Prescribing for Pericoronitis among Dentists: Inappropriate Pericoronitis Treatment Is a Critical Factor of Antibiotic Overuse in Dentistry"

_ijerph, 2021, doi:10.3390/ijerph18136796_

Round 1

Reviewer 1 Report

The authors could provide some description of polymicrobial biofilms in this context, and may be a picture of a retrieved biofilm under a microscope.    

Reviewer 2 Report

Abstract: Please make the conclusions of your study more clear at the end of the abstract.

Graphical abstract: The graphical representation is good and easy to understand.

Introduction: Easy to read and gives a sufficient definition of what pericoronitis is. Authors are critically towards the use of antibiotics in dentistry.

The Terminology Box:

Explains terms as operculitis to readers which are not known by reader which are not dentists.

Etiopathogenesis:

I think that X-ray examination should be standard in cases of pericoronitis and severe tooth inflammation.

Remarks:

Please say it more clearly: What are the main predisposing factors of pericoronitis.

Is there a difference concerning females and males?

Which roles play upper respiratory infections?

Do you find seasonal amassment of pericoronitis?

What about nutrition?

What`s about competitive athletes?

Make it clearer that the overuse of antibiotics may contain considerable risks for further response to needed antibiotics, but keep in mind that antibiotics can be necessary essentials to prevent the patient severe health problems.

The article is easy to read and may be informative for readers which are interested in this theme and are not dentists. However, some parts can be shortened.

Worth for publication after minor corrections.

Reviewer 3 Report

interesting paper but in need of revision

work on grammar and style, many typos

please use a systematic review approach, define the PICO 

literature review, deficient, update with new papers,some suggestions:

Biomed Res Int. 2021 Feb 28;2021:6664434.

Front Microbiol. 2020 Aug 5;11:1888. 

J Family Med Prim Care. 2020 May 31;9(5):2370-2374.

in the Pericoronal tissue surgery (line 438), please include the less invasive approaches such as use of diode lasers and the adjunct use of orthodontics to help alleviate the problem(Compend Contin Educ Dent. 2017 Apr;38(eBook 5):e18-e31.; Borzabadi-Farahani A., Cronshaw M. (2017) Lasers in Orthodontics. In: Coluzzi D., Parker S. (eds) Lasers in Dentistry—Current Concepts. Textbooks in Contemporary Dentistry. Springer, Cham. https://doi.org/10.1007/978-3-319-51944-9_12)

Reviewer 4 Report

Page 4. Line 154 to 155. "The rotation" According to this paragraph the authors describe the relevance of morphology, shape and/or size in pericoronitis. I have some question about it:   1. Could it be possible to review the Winter and Pell and Gregory's classification related to pericoronitis?   2. Could it be possible to make a table explaining the frequency of pericoronitis according to position of third molar?   3. Could it be possible to develop a table related to the percentage of antibiotics most used on pericoronitis treatment?   Figure 1. The figure 1 seems to be the graphical abstract. Please review and correct it. I recommend to explain in the graphical abstract the content and objective of your manuscript. 

Table 1. Could it be possible to explain how the menstruation and stress are involved in the pericoronitis (please explain physiological aspects of menstruation and stress in pericoronitis).

Page 6, Line 222 "Although X-ray". Could it be provided any reference about that? Otherwise, explain in a better manner why authors consider that X-ray is not necessary in the diagnosis of pericoronitis (please include an explanation related to the classification of impacted molar mentioned above).  

Treatment

Line 470 to 471: Why do authors refer to antibiotic overuse? Could the authors give an explanation about it, by considering the obtained data?

What is the opinion of the authors about Photodynamic Therapy in Pericoronitis? Could they give some information about it?  

Overall comments

The manuscript is interesting. The authors explain the main aspects of pericoronitis, focusing on several types of treatments, however, this review seems to be  incomplete. The authors need to explain the classification of pericoronitis, the importance of using antibiotic therapy recommended by the authors. They need to develop tables and modify the abstract figure.

Round 2

Reviewer 3 Report

thank you for the revisions

why did you just include papers January 2000 , update it to 2021

mention in abstract that Systematic review of antibiotic prescribing for pericoronitis, summarize the findings of the systemic revivew in abstract

the 2nd part is a narrative review, mention in abstract

methodology, 1st part defice the PICO properly, explain and identify each part

expaln more on the Questionnaires among dentists and its copartments what was its purpose ?, 

i would suggest you 1st present the 2nd part, then focus on 1st part, 

chang the title as well to reflect that

Reviewer 4 Report

Dear authors the manuscript was improved in this new revision, now is better than previous version, and is more understandable. Now I recommend it for publication without changes.
